# Modelling suggests limited change in the reproduction number from reopening Norwegian kindergartens and schools during the COVID-19 pandemic

Martin Rypdal[1]*, Veronika Rypdal[2,3], Per Kristen Jakobsen[1], Elinor Ytterstad[1], Ola Løvsletten[4], Claus Klingenberg[2,3], Kristoffer Rypdal[1]

1 Department of Mathematics and Statistics, UiT–The Arctic University of Norway, Tromsø, Norway,
2 Department of Clinical Medicine, UiT–The Arctic University of Norway, Tromsø, Norway, 3 Department of Pediatrics, University Hospital of North Norway, Tromsø, Norway, 4 Department of Community Medicine, UiT–The Arctic University of Norway, Tromsø, Norway

* martin.rypdal@uit.no

## Abstract

### Background

To suppress the COVID-19 outbreak, the Norwegian government closed all schools on March 13, 2020. The kindergartens reopened on April 20, and the schools on April 27 and May 11 of 2020. The effect of these measures is largely unknown since the role of children in the spread of the SARS-CoV-2 virus is still unclear. There are only a few studies of school closures as a separate intervention to other social distancing measures, and little research exists on the effect of school opening during a pandemic.

### Objective

This study aimed to model the effect of opening kindergartens and the schools in Norway in terms of a change in the reproduction number ($R$). A secondary objective was to assess if we can use the estimated $R$ after school openings to infer the rates of transmission between children in schools.

### Methods

We used an individual-based model (IBM) to assess the reopening of kindergartens and schools in two Norwegian cities, Oslo, the Norwegian capital, with a population of approximately 680 000, and Tromsø, which is the largest city in Northern Norway, with a population of approximately 75 000. The model uses demographic information and detailed data about the schools in both cities. We carried out an ensemble study to obtain robust results in spite of the considerable uncertainty that remains about the transmission of SARS-CoV-2.

### Results

We found that reopening of Norwegian kindergartens and schools are associated with a change in $R$ of 0.10 (95%CI 0.04–0.16) and 0.14 (95%CI 0.01–0.25) in the two cities under

**Data Availability Statement:** All relevant data are within the manuscript and its Supporting Information files.

**Funding:** The authors received no specific funding for this work.

**Competing interests:** The authors have declared that no competing interests exist.

investigation if the in-school transmission rates for the SARS-CoV-2 virus are equal to what has previously been estimated for influenza pandemics.

## Conclusion

We found only a limited effect of reopening schools on the reproduction number, and we expect the same to hold true in other countries where nonpharmaceutical interventions have suppressed the pandemic. Consequently, current *R*-estimates are insufficiently accurate for determining the transmission rates in schools. For countries that have closed schools, planned interventions, such as the opening of selected schools, can be useful to infer general knowledge about children-to-children transmission of SARS-CoV-2.

## Introduction

In response to the COVID-19 pandemic, the governments of most countries have introduced nonpharmaceutical interventions (NPI) to limit the spread of the virus. The most identifiable NPIs are travel restrictions and social distancing measures, such as isolation for infected individuals and quarantines for their contacts (combined with testing and contact tracing), closure of physical workplaces, and school closures. The effect of these measures is currently a topic of intense research [1, 2]. In particular, the effect of school closures is mostly unknown [3]. By early April 2020, 188 countries had closed schools countrywide, and more than 90% of the world's learners were affected [4].

On March 12, the Norwegian government announced a series of restrictive infection control measures, including school closure from March 13. However, due to a situation with few COVID-19 cases in Norway, in particular among children and the general disruptive effects of school closures, the government announced already mid-April, a gradual reopening starting with kindergartens from April 20, schools for children born later than 2013 from April 27, and schools for children born later than 2009 from May 11. The Norwegian Institute of Public Health (NIPH) and the Ministry of Education and Research have published guidelines for infection prevention and control in schools during the COVID-19 pandemic [5]. The reopening of schools on April 20 and April 27 were not accompanied by relaxation of other control measures until later in the spring of 2020.

Based on the initial outbreak in China, only 1% of confirmed cases were children aged 1–9 years [6], and a similar proportion in this age group (1.4%) is reported in Norway [7]. Due to milder disease in children, we expect the proportion of COVID-19 cases ascertained to be significantly lower for children than adults, and several studies assume a uniform attack rate among age groups [8, 9]. A uniform attack rate would imply that testing in China only ascertained around 10% of cases in children because asymptomatic children are likely not tested. Similar proportions would be expected in Scandinavian countries, depending on testing regimes. If a large proportion of the infections in children are unrecorded, it is difficult to conclude on the effect of school closures based on the currently available data.

One approach, presented in the 13th report on the COVID-19 virus of the Imperial College COVID-19 Response Team [2], is to estimate *R* as a function of time in different countries, and follow the evolution of this number as countries have put in place various interventions. The study from Imperial College analyzed 11 European countries and found that school closures have the effect of reducing the *R* by up to 50%, with a best estimate of 20–25%. The authors argued that there is considerable uncertainty regarding these numbers due to the short

time between different interventions (schools closing, social distancing, banning of public events, self-isolation, and lockdown). However, these results are consistent with previous analyses comparing the transmission of infectious diseases on weekdays and weekends [10]. In a recent study of the COVID-19 outbreaks in Wuhan and Shanghai, Zhang *et al*. use contact surveys to set up contact matrices in an age-structured Susceptible-Infectious-Recovered (SIR) model, and found, in simulations, that school closures can reduce the peak incidence by 40–60%, but are alone, not sufficient to interrupt the transmission [11]. The results of Zhang *et al*. are consistent with evidence from the 1918 influenza pandemic, suggesting that school closures could have reduced the total number of infections by 15%, and the peak attack rates by 40% [12–16].

Given the uncertainty of disease spread in schools, and the effect of keeping schools closed, we must evaluate decisions to reopen schools based on the state of the disease development in the actual country, region, and city. The decision to reopen schools in Norway followed an announcement on April 6 [17], where the NIPH, presented an estimated $R$ of 0.70 for Norway, with a 95% confidence interval (95%CI) of 0.45–1.

In this study we aimed to model the effects on $R$ after school opening in two Norwegian cities. We performed an ensemble of experiments in an individual-based model (IBM) that is similar to the one used by Ferguson *et al*. [18–20] and other complex transmission models [21–25]. There exist similar IBM studies that focus on the effects of NPIs on the spread of SARS-CoV-2 [26]. However, as far as we know, no other studies that evaluated a controlled school reopening using this methodology were published when this work was carried out in the spring of 2020. However, the field is evolving rapidly, and in August, an IBM study on strategy for reopening schools in the UK was published [27]. We use a novel approach to infer results that remain robust under the uncertainties about SARS-CoV-2 virus dynamics.

## Methods

The method used in this paper consists of two main parts. The first part is to set up and run experiments in an IBM for each of the two cities we investigate. The second part is to estimate the change, and the uncertainty, in the effective reproduction number resulting from school opening.

The model setup consists of two parts. First, we needed to construct the network on which we simulate transmission, and secondly, we needed to specify the transmission model.

Constructing the transmission network was a non-trivial task. Since we aimed to evaluate the effect of school opening, we needed to include schools in the network. We also needed to include families in the network since the families tie together different schools. For instance, if two siblings attend two different schools, their family represents a connection between the two schools that can cause an outbreak to spread from one school to another. Our approach was first to construct families randomly using probability distributions consistent with the demographic data for each city. The random assignment of schools used probabilities derived from the municipalities' data for the number of students in each school. If two children in the same family both attended middle school, they were assigned to the same school. The same was true for elementary schools and kindergartens. Before high school, Norwegian children attend schools in their neighborhoods. To capture this structure, we first assigned children to middle schools and then distributed younger siblings (and younger children without older siblings) to elementary schools and kindergartens in middle schools' neighborhoods.

Once the networks were constructed, we modeled disease transmission using a standard probabilistic IBM with three layers. In this model, each individual's probability of being infected was proportional to a weighted sum over all infectious individuals in the city. We

used one weight for family members, one for schoolmates, and one for other individuals that the person could encounter. The transmission rates in families, in schools, and for other encounters were the most critical parameters in the model.

We ran model experiments with varying parameters, and for each parameter set, we carried out two experiments. One where schools remained closed, i.e., the transmission rates in schools were set to zero, and a second experiment where the schools were opened. We estimated the effective reproduction number for each pair of runs and compared the value with schools closed to the value estimated from the run where the schools were opened. To estimate the reproduction number, we fitted the data from the model runs to a susceptible-exposed-infected-recovery (SEIR) model. The rationale for using an SEIR model was that this technique resembles the method used to assess Norway's reproduction number at the time.

The final part of our analysis was to analyze the change in the reproduction number over the set of model experiments. We focused on the dependence between the difference in the reproduction number (between schools closed and schools opened) and the transmission rate in schools, which was the most critical and most uncertain transmission model.

The details of our methodology are described below.

## COVID-19 outbreaks in Oslo and Tromsø

Oslo is the largest city and the epicenter of the COVID-19 outbreak in Norway. By May 22, the city had 3.73 confirmed cases per 1000 inhabitants, compared to 1.54 per 1000 in Norway as a whole. There were 2.18 cases per 1000 inhabitants by May 22 in the city of Tromsø.

## Individual-based model (IBM) of school opening

The model used detailed demographic data for the two cities, statistics for Norwegian families, and detailed information about the network of city schools and kindergartens. We made a wide range of different assumptions about the transmission rates within and outside families and schools. The experiments carried out conformed with the Norwegian government's plan for school reopening and were evaluated by comparison with model simulations where the schools remain closed.

We built networks of families and schools that were consistent with demographic data and school data collected from Statistics Norway and the municipalities of the two cities [28, 29]. In the first step of the construction, we associated every infected individual with a family. Every child (under the age of 18 years of age) was associated with a kindergarten or a school. Consistent with demographic data, the number of families was 0.45 times the total population, 24% of families were couples without children, 18.5% of families comprised of two parents with children, and 7% of families were composed of one parent with children. For families with children, the average number of children was 1.75.

The age of each child was drawn randomly according to the age distribution in each city. We assigned each family with children to one middle school, one elementary school, one high school, and one kindergarten. The middle schools and high schools were drawn randomly with probabilities proportional to the school sizes. The elementary schools were drawn randomly from the set of elementary schools associated with the middle school assigned to the family. Our model associated each elementary school with ten different kindergartens and assigned each family to a kindergarten associated with the family's elementary school. Depending on their age, we assigned children to their family's kindergartens, elementary school, middle school, or high school.

We modeled transmission in the networks stochastically. In each time step, we found the probability of infection for each individual by computing a force of infection for this individual.

A weighted sum over three terms defined the force of infection. The first term was a weighted sum over infectious persons in each individual's family, the second over the infectious persons in each individual's school, and the third term over all infectious individuals in the city.

More precisely, in each time step $t \rightarrow t+\Delta t$, a susceptible individual $i$ could be infected with probability $p_i = 1 - e^{-\lambda_i \Delta t}$, where the force of infection

$$\lambda_i = \lambda_i^{(1)} + \lambda_i^{(2)} + \lambda_i^{(3)}$$

varied with time. The first term, $\lambda_i^{(1)}$, describes the force of infection within the household:

$$\lambda_i^{(1)} = \frac{\beta_1}{n_i^\alpha} \sum_{j \text{ in family of } i} J_j \rho_j \left(1 + c_j\right) \kappa \left(t - \tau_j - s_j\right).$$

The second term, $\lambda_i^{(2)}$, describes the force of infection within schools:

$$\lambda_i^{(2)} = \frac{\beta_2}{m_i} \sum_{j \text{ in school of } i} J_j \rho_j \left(1 + (2\Psi - 1)c_j\right) \kappa \left(t - \tau_j - s_j\right).$$

The third term, $\lambda_i^{(3)}$, describes the force of infection from random encounters in society:

$$\lambda_i^{(3)} = \frac{\beta_3}{N} \sum_{j \neq i} J_j \rho_j \left(1 + c_j\right) \kappa \left(t - \tau_j - s_j\right).$$

Here, $J_j = 1$ if individual $j$ is infected, and $J_j = 0$ otherwise.

The set-up closely follows what was used by Ferguson *et al*. [18]. The number $\tau_j$ denotes the time at which individual $j$ was infected, $s_j$ is the incubation time, and $\kappa(t)$ describes how the infectiousness decreases with time as individuals go from infected to recovered/dead. We distinguished between serious infections ($c_j = 1$) and non-serious infections ($c_j = 0$) and implemented lower probability of serious infection for children than for adults. The seriousness of an infection affects the infectiousness of individuals by a factor $1+c_j$, but also reduces infectiousness in schools by a factor $\Psi$ if seriously infected. The latter was meant to model the effect of symptomatic children staying home from school/kindergarten. The relative infectiousness of individuals, $\rho_i$, was randomly chosen for every individual in the population. The number of family members in the family of individual $i$ is denoted $n_i$. The parameter $\alpha$ describes how the in-household transmission scales with the size of families. The number of students in the school of individual $i$ is $m_i$, and $N$ is the total population. The model was run with a time step of $\Delta t = 0.25$ days. The parameter values we used are shown in S1 File.

## Model runs and analyses

In the IBM we carried out a number of simulations with transmission coefficients ($\beta_1$, $\beta_2$, and $\beta_3$), each drawn randomly from uniform distributions on the interval 0.2–1. In the model experiments, all schools and kindergartens were closed during the first 14 days of the simulation ($\beta_2 = 0$). These 14 days correspond to the period from April 6 to April 20. During this period all kindergartens and schools in Norway were closed. Following the actual implementation of school reopening, we opened kindergartens (children born 2014 or later) after running the models for 14 days and grades 1–4 (children born 2010–2013) opened after 21 days. The reopening of schools for older children coincided with other changes in restrictions and was not included in this modelling study.

In the model, school openings are implemented by including the term $\lambda_i^{(2)}$ in the force of infection. The number of infectious individuals in the start of the simulations were taken to be

5000 and 300 in Oslo and Tromsø, respectively. This was roughly three times the number of confirmed cases in each city (at the time the simulations were set up) in order to account for unrecorded cases. Since our results concern changes in the reproductive number, and not the absolute number of cases, they are largely insensitive to the initial conditions. For the city of Oslo, we carried out 121 pairs of model experiments, and for the city of Tromsø we carried out 295 pairs. The two runs in a pair were identical for the first 14 days, and after that they branched in two directions: one run where schools opened, and a second run with the same parameters, that continued with schools closed. Model parameters were resampled for each pair.

We estimated $R$-values by fitting the number of infectious individuals to $I(t)$ in simulations of a susceptible-exposed-infected-recovery (SEIR) model.

In the IBM we quantified the effect of school openings as the difference $\Delta R$ between the estimated $R$-values in two simulations where schools were opened in one, and not in the other, but otherwise identical. Clearly, $\Delta R$ depended on the in-school transmission rate $\beta_2$. For IBMs of the type used here, the transmission rates estimated by Ferguson *et al*. [18] are reference values that have also been used in other modelling studies [30]. Ferguson *et al*. found the in-school transmission rate $\beta_2 = 0.94$ per/day to match well with attack rates for children during the 1957–1958 influenza pandemic [31]. Based on this, we define a dimensionless relative in-school transmission rate $r = \beta_2/(0.94 \text{ per day})$. The factor $r$ characterizes the in-school transmissibility of the SARS-CoV-2 compared to a pandemic influenza, i.e., $r = 1$ corresponds to a pandemic influenza. We varied $\beta_2$ in the range from 0.2 per day to 1.0 per day, i.e., $r$ between 0.21 and 1.06. The transmission rates $\beta_1$ and $\beta_3$ were also chosen from the uniform distribution on the interval (0.2,1).

## Estimation of reproduction number using an SEIR model

To estimate the basic reproduction number in Norway we used a one-population version of the SEIR model used by the NIPH [17], and our estimation of $R$ follows their approach (See S1 File for details).

The method was to assume fixed parameters (except $R$) and assume a stepwise time evolution of the reproduction number, with one constant value, $R^{(0)}$, before March 15, a second constant value, $R^{(1)}$, between March 15 and April 20, and a third value, $R^{(2)}$, after April 20. The transitions were smoothed using the hyperbolic tangent function and two characteristic transition times that were estimated as a part of the procedure. Varying these five parameters, we minimized the square error (on a linear scale) between the time series of hospitalized patients with COVID-19 in Norway and the predicted number of hospitalized patients from the SEIR model. The NIPH has published the SEIR-parameters, the assumptions on the proportion of symptomatic individuals that require hospitalization, and assumptions on the length of hospitalizations [17]. The NIPH employs a metapopulation-version of the SEIR model, where each municipality in Norway is modelled separately and coupled via contact a matrix estimated from cellphone data provided by the telecommunication company Telenor, but we used the one-population version of the model.

Uncertainty was estimated by repeated addition of normal distributed noise to the hospitalization data and re-estimation of parameters. We used a noise with a standard deviation twice as large as the sample standard deviation of the difference between the optimal fit and the hospitalization data. Data on the number of hospitalized patients with COVID-19 was downloaded from The Norwegian Directorate of Health [5] on May 22. We note that number of hospitalized COVID-19 patients in Norway never exceeded 324, so limited hospital capacity did not influence the number of hospitalizations.

As a supplement to this method, we also applied a simple non-parametric method to the time series of confirmed infections in Norway, as well as in the cities of Oslo and Tromsø. The approach, which is described in the S1 File, is similar to the method presented by Thomson *et al.* [32].

## Results

Fig 1 shows the development of the accumulated number of infections for selected sets of $\beta$-parameters in our ensemble of simulations. The black curves represent simulations for a situation without school opening ($\beta_2 = 0$), and red curves with school opening. The difference between the red and black curves, which represents the effect of school opening, is small for all examples except for the one shown in panel B. The reason for this is that $R$, after school opening, is less than unity for panels A, C, and D. For these examples, the epidemic decays even after the opening of school, and hence the opening has little consequence. For the example shown in panel B, however, the reproduction number $R$ is less than unity without opening, and $R+\Delta R$ greater than unity with opening of schools. This is an example where school opening leads to a second wave of infection.

This risk of a new wave of infections depends on the magnitude of $R$ prior to school opening. As will be shown below, we found that if in-school transmission rates for the SARS-CoV-2 virus are similar to what is believed to be the case for influenza pandemics ($r = 1$), we estimated

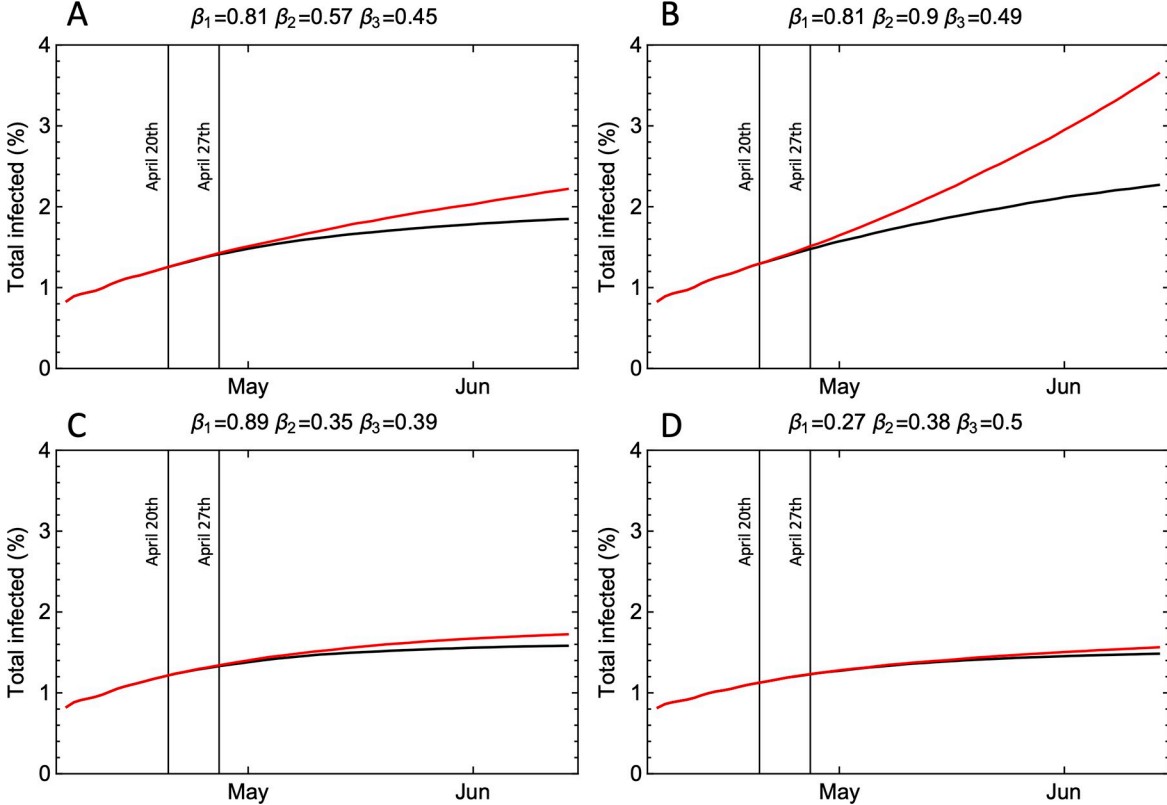

**Fig 1. The total number of infected individuals in model experiments for the city of Oslo.** The black curves are simulations where schools remain closed and the red curves are simulations where the schools are opened in two steps (April 20th and April 27th). A: Transmission rates $\beta_1 = 0.81$ per day (households), $\beta_2 = 0.57$ per day (schools), and $\beta_3 = 0.45$ per day (other contacts). B: Transmission rates $\beta_1 = 0.81$ per day, $\beta_2 = 0.90$ per day, and $\beta_3 = 0.49$ per day. C: Transmission rates $\beta_1 = 0.89$ per day, $\beta_2 = 0.35$ per day, and $\beta_3 = 0.39$ per day. D: Transmission rates $\beta_1 = 0.27$ per day, $\beta_2 = 0.35$ per day, and $\beta_3 = 0.5$ per day.

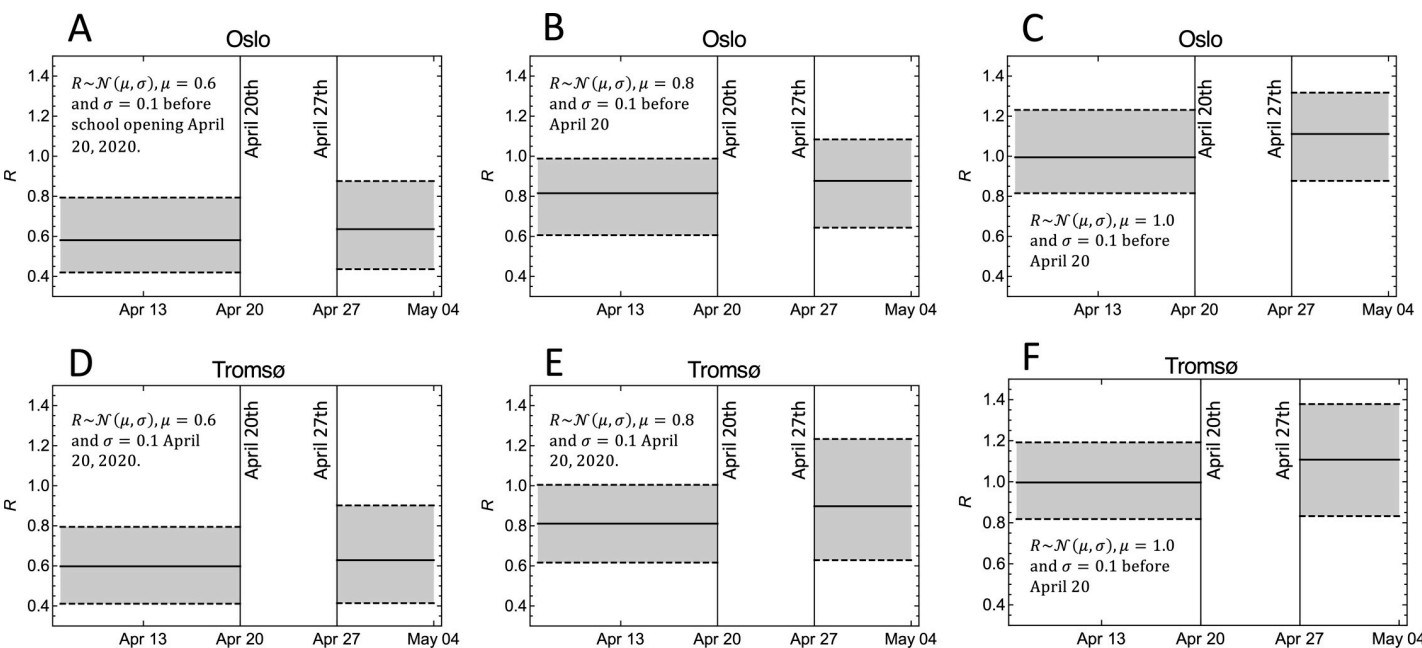

**Fig 2. The effect of school opening on the estimates of the basic reproduction number *R* under different assumptions on the value before school opening.** The solid lines are the weighted medians, and the dashed lines show weighted 95% CI. A: The model ensemble for the city of Oslo weighted by a normal distribution with mean 0.6 and standard deviation 0.1 for the reproduction number prior to April 20. B: As (A) but weighted by a normal distribution with mean 0.8 and standard deviation 0.1. C: As (A) and (B) but weighted by a normal distribution with mean 1.0 and standard deviation 0.1. D, E, and F: As (A-C), but for the city of Tromsø.

for Oslo Δ*R* = 0.10 (95%CI 0.04–0.16) and for Tromsø, Δ*R* = 0.14 (95%CI 0.01–0.25). This means that the most interesting range of *R* prior to school opening to explore is the interval 0.6–1.0. The result of such an exploration is shown in Fig 2. The three panels show the estimated *R* before April 20 and after April 27, assuming different values of *R* before April 20. The way we were able to assume different initial values of *R*, was that we weighted simulations by their estimated *R* values before April 20 using gaussian weight distributions centered around *R* = 0.6, 0.8, and 1.0, respectively, with standard deviation 0.1. To the right in each panel, the *R*-values after school opening is plotted. The most interesting case is probably *R* centered around 0.8 (panels B and E), because this is close to the most probable *R* estimated for the two cities (see S1 Fig), and in this case, the weighted sum over model runs with *R* greater than 1 after April 27 gave a probability of 0.13 for having *R*+Δ*R*>1 after school opening in Oslo. In Tromsø, this probability is somewhat higher, 0.28, as shown in panel E. Since the magnitude of *R* prior to school opening is only known with considerable uncertainty (S1 Fig), these probabilities are also associated with great uncertainty, but their magnitudes still provide an indication of the risk for a second wave of infection associated with school openings.

The direct effect on the additional cases of infection in the two cities eight weeks after school opening is shown in S2 Fig. The figure demonstrates the importance of the magnitude of *R* prior to opening.

The variation in Δ*R* in Fig 2 was mostly determined by variation in the relative in-school transmission rate *r*. We found the relationship between Δ*R* and *r* to be approximately linear over the set of simulations with different parameter, and for both cities. We estimated Δ*R* = 0.10 *r* for Oslo, and Δ*R* = 0.14 *r* for Tromsø (Fig 3 and S3 Fig). The standard errors were 0.004 for 0.10, and 0.005 for 0.14. It is on the basis of these linear relationships that we found Δ*R* = 0.10 and Δ*R* = 0.14, assuming that in-school transmission rates are similar to what is believed to be the case for influenza pandemics (*r* = 1). If in-school transmission rates are 50%

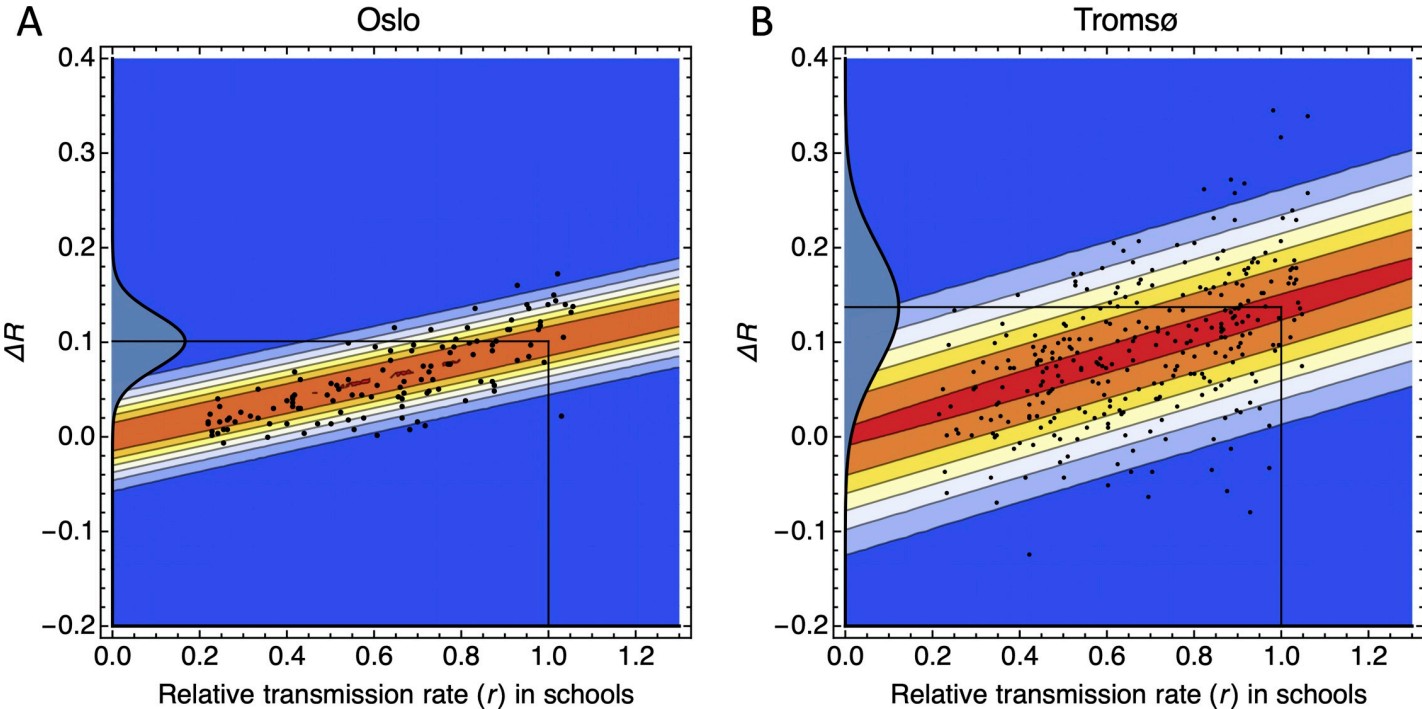

**Fig 3. The change in reproduction number ΔR plotted against the relative transmission rate r.** The value $r = 1$ corresponds to the transmission rate 0.94 per day, which is taken as a rough estimate for the in-school transmission rate during an influenza pandemic [18]. The black points are model simulations for randomly selected $\beta$-parameters, and the contours show the conditional probability density $p(\Delta R|r)$ estimated using the method in [33]. A: For the city of Oslo. B: For the city of Tromsø. The probability densities on the vertical axes are $p(\Delta R|r = 1)$.

of the estimate for influenza pandemics, the estimated changes in the reproduction number are $\Delta R = 0.05$ and $\Delta R = 0.07$ for the two cities, respectively.

Another objective of this study was to assess if we can use the estimated $R$ after school openings to infer the rates of transmission between children in schools. The simulations show that we are not able to make such inference. The main reason for this is that the linear relationships between $\Delta R$ and $r$ have small proportionality factors (0.10 and 0.14), implying that uncertainty of in-school transmission rates is amplified relative to uncertainty in $\Delta R$ (.). In order to constrain in-school transmission rates from the change (or lack of change) in $R$ in Norway, we would need to estimate $\Delta R$ with significantly higher accuracy than we can obtain from currently available hospitalization data using SEIR models (Fig 4), or from non-parametric estimation methods of $R$ (S1 Fig).

## Discussion

For the COVID-19 pandemic, early evidence from the outbreak in China showed that children of all ages are susceptible, but that disease, in general, is milder in children than for adults [34]. There is currently little concrete data on how the attack rate varies with age, and how common asymptomatic infections are in different age groups. Recent data from Wuhan shows that the secondary attack rate in households was as high for children as adults [9], but it is still unclear to what extent children pass on the virus, and consequently, the children-to-children transmission rate is unknown [35–37]. It can be argued that if children have milder symptoms than adults, they may also be less infectious, but this is unclear at this point [38]. Children with mild symptoms are less likely to be absent from school while they are infectious, and it may be more

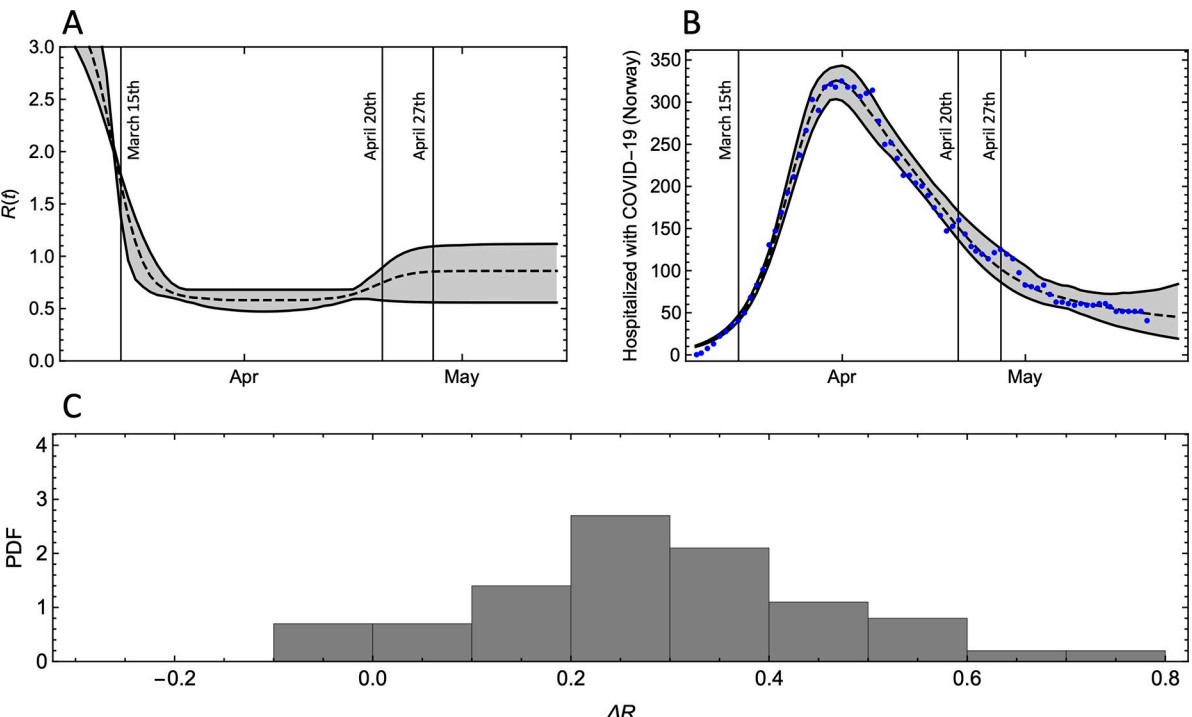

**Fig 4.** A: Estimate of the reproduction number in Norway under the assumption that it follows a stepwise constant trajectory with smooth steps. B: The number of hospitalized COVID-19-pateinets in Norway (blue points) and the model results (median and 95% CI) for the estimated $R$-curve shown in (A) (See S1 File for details). C: The distribution of the estimated change in reproduction number after April 20.

difficult for children to conform to strict hygiene measures and social distancing. Our findings suggest that it is challenging to infer these data through assessments of countrywide school openings, even if one makes the unrealistic assumption that the change in $R$ after April 20 is not influenced by other factors than schools opening, such as changes in behavior in other age groups.

Our model results suggest that the controlled opening of schools in Norway will lead to a change in $R$ of less than 0.25, and most likely in the range 0.10 to 0.14. This holds true despite the uncertainty about the role of children in the spread of the SARS-CoV-2 virus. Since Norway had strongly suppressed the COVID-19 outbreak by the middle of March, a change in $R$ of the order of 0.10 to 0.14 will not have had a strong effect on the number of infections.

A strength of our study is that the estimates are robust under a wide range of assumptions about transmission rates and reflect changes in the contact networks associated with school openings. We model these network changes using detailed data about the actual school structure in the two Norwegian cities. This aspect of the modeling is carried out in greater detail in this work than in the previous modeling of influenza pandemics [18–20].

There is uncertainty in the reference value for in-school transmission rates [18], but our results show low sensitivity to the transmission rates. The model results for the city of Oslo show a change in the $R$ less than 0.25, even for in-school transmission rates which is 20% higher than the influenza reference value. Moreover, there is evidence that a significant proportion of children continue to mix with other children after unplanned school closures [39]. If we had included this effect, it would reduce the estimated effect of school openings. On the other hand, there are also trickle-down effects of school openings, such as increased contacts between adults (teachers and parents) that we have not taken into account.

Based on our modeling results, we conclude that controlled reopening of schools in countries that have the first wave of the COVID-19 pandemic suppressed will only have a limited effect on the reproduction number. The benefits to children's health of opening schools are well-documented [35, 36].

## Supporting information

**S1 File. Model parameters and methods for estimation of *R*.**
(DOCX)

**S1 Fig.** A: Estimate of *R* based on confirmed cases in Norway. B: Estimate of *R* for Oslo. C: Estimate of *R* in Tromsø.
(TIFF)

**S2 Fig. Shows the difference in the total number of cases with schools opened and with schools closed, eight weeks after school opening.** The differences in the number of cases is plotted against the *R*-value before school opened (April 20). Each point represents a model simulation. A: Simulations for the city of Oslo. B: The city of Tromsø.
(TIFF)

**S3 Fig. This figure shows the same data as Fig 3 in the manuscript, but the axes are changed so that *r* is plotted as a function of $\Delta R$.** The black points are model simulations for randomly selected $\beta$-parameters, and the contours show the conditional probability density $p(\Delta R|r)$ estimated using the method in [7]. The probability densities on the axes are included to illustrate how the uncertainties are amplified. On the $\Delta R$-axes we have chosen normal distributions $p(\Delta R)$ with standard deviation 0.02, and on the *r*-axes we show the corresponding distributions $p(r)$ obtained from integration of $p(\Delta R|r)p(\Delta R)$ over $\Delta R$. A: For the city of Oslo. B: For the city of Tromsø.
(TIFF)

## Author Contributions

**Conceptualization:** Martin Rypdal, Veronika Rypdal, Claus Klingenberg, Kristoffer Rypdal.

**Formal analysis:** Per Kristen Jakobsen, Elinor Ytterstad, Ola Løvsletten, Kristoffer Rypdal.

**Investigation:** Martin Rypdal, Claus Klingenberg.

**Methodology:** Martin Rypdal, Per Kristen Jakobsen, Elinor Ytterstad, Kristoffer Rypdal.

**Project administration:** Martin Rypdal.

**Software:** Martin Rypdal, Ola Løvsletten.

**Visualization:** Martin Rypdal.

**Writing – original draft:** Martin Rypdal, Veronika Rypdal, Claus Klingenberg.

**Writing – review & editing:** Martin Rypdal, Veronika Rypdal, Per Kristen Jakobsen, Elinor Ytterstad, Ola Løvsletten, Claus Klingenberg, Kristoffer Rypdal.

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
