## [Decision Letter · Decision Letter 0]

27 Nov 2020

PONE-D-20-25739

Modelling Suggests Limited Change in the Reproduction Number from Reopening Norwegian Kindergartens and Schools During the COVID-19 Pandemic

PLOS ONE

Dear Dr. Rypdal,

Thank you for submitting your manuscript to PLOS ONE. After careful consideration, we feel that it has merit but does not fully meet PLOS ONE’s publication criteria as it currently stands. Therefore, we invite you to submit a revised version of the manuscript that addresses the points raised during the review process.

The Authors are expected to address all the criticisms by all Reviewers. In particular, please provide more details on the method/model/parameters (Reviewers #1, 2 & 3), clarify potential behavioral change during close closure or being sick and clarify/use consistent notations for the school types (Reviewer #1), clarify the use of cellphone data for estimating the contact matrix (Reviewer #2), reassess the suitability of the influenza-based transmission model for COVID-19 (Reviewers #3). In additional to the above comments, please address,

Please clarify if the maximum hospital capacity has been reached during the study period, so some COVID-19 patients were actually not admitted to hospital.Besides reopening of schools, were there any other major relaxation of control measures in the general population that would have affected the hospitalization data?Supp Table 1. Please add the definition of the parameters.

We look forward to receiving your revised manuscript.

Kind regards,

Eric HY Lau, Ph.D.

Academic Editor

PLOS ONE

Journal Requirements:

2.Thank you for stating the following financial disclosure:

 [No].

3. Your abstract cannot contain citations. Please only include citations in the body text of the manuscript, and ensure that they remain in ascending numerical order on first mention.

Additional Editor Comments (if provided):

The Authors are expected to address all the criticisms by all Reviewers. In particular, please provide more details on the method/model/parameters (Reviewers #1, 2 & 3), clarify potential behavioral change during close closure or being sick and clarify/use consistent notations for the school types (Reviewer #1), clarify the use of cellphone data for estimating the contact matrix (Reviewer #2), reassess the suitability of the influenza-based transmission model for COVID-19 (Reviewers #3). In additional to the above comments, please address,

1. Please clarify if the maximum hospital capacity has been reached during the study period, so some COVID-19 patients were actually not admitted to hospital.

2. Besides reopening of schools, were there any other major relaxation of control measures in the general population that would have affected the hospitalization data?

3. Supp Table 1. Please add the definition of the parameters.

Reviewers' comments:

Reviewer's Responses to Questions

**Comments to the Author**

1. Is the manuscript technically sound, and do the data support the conclusions?

Reviewer #1: Yes

Reviewer #2: Partly

Reviewer #3: Yes

2. Has the statistical analysis been performed appropriately and rigorously? 

Reviewer #1: Yes

Reviewer #2: I Don't Know

Reviewer #3: Yes

3. Have the authors made all data underlying the findings in their manuscript fully available?

Reviewer #1: No

Reviewer #2: No

Reviewer #3: No

4. Is the manuscript presented in an intelligible fashion and written in standard English?

Reviewer #1: No

Reviewer #2: No

Reviewer #3: Yes

5. Review Comments to the Author

Reviewer #1: This manuscript uses an individual based model incorporating household, school and community mixing to estimate the impact on R (reproduction number) of school reopening in Norway (Oslo & Tromso). The approach taken seems sound, and does a good job of capturing the implications of parameter uncertainty. The limitations arising from this (and from availability of calibration data) are recognised.

My one main suggestion is that I found some of the methodology unnecessarily difficult to follow. Each of the individual components is reasonably described, but how they fit together (and how they relate to the specific questions addressed) could be more clearly described.

Further comments below:

please number pages and lines!

check spelling & punctuation: "serios" somewhere, and misplaced commas.

Introduction:

"no other study uses as detailed data as we have collected in our experiments" Are you referring to empirical data or model outputs here?

Obviously this is a very fast moving area, one further study I am aware of (though not involved with) that has appeared since is https://doi.org/10.1016/S2352-4642(20)30250-9

Methods:

Seriously infected children make a reduced contribution to the school-based force of infection, based on the assumption that they will stay home from school. Do these children make an increased contribution to the home-based force of infection (where they are now spending more time)?

Do you assume any change in behaviour (eg, isolation) relating to development of symptomatic/severe disease in other age groups? How might this influence your interpretation of model behaviour?

The timing of school opening and closures is a unclear: "In the model experiments, schools remain closed for 14 days (β 2 = 0), and then kindergartens opened. Grades 1-4 opened one week later." When schools closed, was this all schools, including kindergartens? After Grades 1-4 opened, did higher grades also open? It may be useful when describing model to associate which school years / age groups / birth years are associated with which type of school, and use consistent terminology throughout.

How were the numbers of experiments (121 for Oslo and 295 for Tromso) chosen? Were these "experiments" indepentent stochastic realisations using the same parameter values, or were the parameter values resampled for every experiment? For the comparison results, did each of these experiments involve running the simulation twice (ie, with and without school opening)? Please explain this more clearly.

I am a bit confused by the description of model runs. In the first paragraph it suggests that you sampled each of the three beta parameters from uniform distributions. The followin paragraph suggests that beta_2 was fixed to 0.94 following Ferguson's models.

Am I understanding correctly that the sole purpose of the compartmental model was to enable estimation of R from hospitalisation data, that the IBM could then be fit to? It would be helpful to explain this more clearly, as it took some piecing together.

Figure 2: it would be helpful to label the rows / columns with identifying factors (eg, city / weight parameters) rather than relying on the caption.

Supp mat: references to Eurosurveillance would need to be updated.

Reviewer #2: thank you for the attempt to investigate the effectiveness of closing of schools during this pandemic. First, it is best to carefully check your English - there exists some spelling mistakes and grammatical errors; also, please check on how to cite reference when quoting. Second, you have mentioned on page 9 that you have used "ensembles of model experiments" of which I could not find the discussions on this. Can you clarify? Third, will it be possible to inform on what basis does the parameter c were assigned such values? also, can you justify the assigned values for psi and alpha? Fourth, on the R estimation, it was not clear why were there two sets of systems of equations, one on SEI and the other on SEIR ? Are the lamda different from the SEIR model to the ones defined on the main text? and, which of the parameters that you have introduced may be in relation to the social/physical distancing? Also, will it be possible to explain further on the contact matrix estimated from the cellphone data. Why can't contact matrix from Prem et al be used? Finally, would you say that your findings differs from that of Zhang's ?

Reviewer #3: Re: Modelling suggests limited change in the reproduction number from re-opening Norwegian Kindergartens and schools during COVID-19

In the manuscripts, the authors considered the effect of kindergartens and schools opening on COVID-19 dynamics in Norway. Both basic and effective reproduction numbers were estimated relative schools openings via the individual-based model (IBM). The found little relative change in reproduction number from opening the schools and concluded that controlled school opening might not alter the cause of progress made on COVID-19 in many countries considering this option. The methodology and results are well-written and supported by good discussion and conclusion. In general, the manuscript is well-organized, and my comments on the work are below:

Major comments:

The authors follow the stochastic model for influenza directly in [1] for COVID-19. The dynamics of COVID-19 is entirely differently from influenza [2]. I can see that some of the parameters are updated for COVID-19, but pre-symptomatic transmission and asymptomatic are two of the factors that make COVID-19 unique. These are not accounted for in this modelling. Thus, I wonder if this stochastic simulation captures the essential dynamics of COVID-19.

It is has been shown that there exist age-dependent susceptibility and infectivity in COVID-19 dynamics [3]. More of this assertion comes from a population with more younger age group than older age group vs with a population with more older age group than younger age group. The Norway age-structure is pyramid [4], meaning there are fewer young children than adults. Thus, allowing contacts between children and adults via school re-opening may not significantly impact COVID-19. However, all these need to be adjusted for in the modelling. How does infectiousness and susceptibility of younger children differ from adults in this modelling? Note, your c_j and ρ_j may answer relative infectiousness but not relative susceptibility.

As mention in 1 above, COVID-19 has its dynamics. Thus, a SIR or SEIR compartmental like models may be over-simplification of reality. The authors did not provide the compartmental like nature of their IBM. Hence, it is easier to assume it is SIR. Also, what is the essence of using SEIR model to estimate R_0? The IBM model can be fitted to data to estimate this. How does the dynamics of your SEEIIR model differ from IBM. Since Age is considered, I guessed age-structured compartmental model with appropriate contact matrix would be closed the approximation of the IBM model.

The results are a little bit difficult to follow. For instance, “ For each city, the figures are based on the whole ensemble of simulations but weighted by a probability density p(R) for the reproduction number prior to April 20,…”. p(R) is not defined anywhere else, even in the supplementary! I may miss this.

Similar to 4 above, “ the Gaussian weight distribution is centred around R=0.6, 0.8 and 1.0, …”. Which Gaussian distribution? All you have in figure 2 are straight lines of the basic reproduction numbers. “In panel B, the probability of having R+ΔR>1 after school opening in Oslo is estimated be 0.13…”. How? Nothing in the methodology pointed to this, and even in panel E, figure 2, what you have are straight lines. Maybe a better figure that depicts these changes will be better.

Minor comments:

Change “serios” to “serious”

“Form” to “are” in the model runs and analyses section

References:

Ferguson, N. M., Cummings, D. A., Cauchemez, S., Fraser, C., Riley, S., Meeyai, A., ... & Burke, D. S. (2005). Strategies for containing an emerging influenza pandemic in Southeast Asia. Nature, 437(7056), 209-214.

https://www.who.int/news-room/q-a-detail/coronavirus-disease-covid-19-similarities-and-differences-with-influenza

Davies, N. G., Klepac, P., Liu, Y., Prem, K., Jit, M., Eggo, R. M., & CMMID COVID-19 working group. (2020). Age-dependent effects in the transmission and control of COVID-19 epidemics. MedRxiv.

https://www.populationpyramid.net/norway/2019/

6. PLOS authors have the option to publish the peer review history of their article (what does this mean?). If published, this will include your full peer review and any attached files.

Reviewer #1: No

Reviewer #2: No

Reviewer #3: No

---

## [Author Response · Author response to Decision Letter 0]

9 Dec 2020

Response to reviewers’ and editor’s comments. In the following we respond point-by-point to the comments of the editor and the reviewers. 

Editor Comments:

Please provide more details on the method/model/parameters (Reviewers #1, 2 & 3). 

Clarify potential behavioral change during close closure or being sick and clarify/use consistent notations for the school types (Reviewer #1). 

Clarify the use of cellphone data for estimating the contact matrix (Reviewer #2)

Reassess the suitability of the influenza-based transmission model for COVID-19 (Reviewers #3). 

We have expanded the descriptions of the methods and made modifications based on the specific comments raised by the three reviewers. See the response to each reviewer for the details. 

In additional to the above comments, please address,

1. Please clarify if the maximum hospital capacity has been reached during the study period, so some COVID-19 patients were actually not admitted to hospital.

This was not the case in Norway. We have added the following sentence to clarify this (Line 328 in revised manuscript): 

“We note that number of hospitalized COVID-19 patients in Norway never exceeded 324, so limited hospital capacity did not influence the number of hospitalizations.” 

2. Besides reopening of schools, were there any other major relaxation of control measures in the general population that would have affected the hospitalization data?

There were not, and this is the reason our study focused on the two first steps of the school reopening. The opening of high school, for instance, was done at the same time as other measures were relaxed. To clarify this, we added the following sentence to the introduction (Line 61 in the revised manuscript): 

“The reopening of schools on April 20 and April 27 were not accompanied by relaxation of other control measures until later in the spring of 2020. “ 

3. Supp Table 1. Please add the definition of the parameters.

The table is updated with definitions. We also included the transmission rates for clarity. 

Reviewer #1: 

This manuscript uses an individual based model incorporating household, school and community mixing to estimate the impact on R (reproduction number) of school reopening in Norway (Oslo & Tromso). The approach taken seems sound, and does a good job of capturing the implications of parameter uncertainty. The limitations arising from this (and from availability of calibration data) are recognised.

My one main suggestion is that I found some of the methodology unnecessarily difficult to follow. Each of the individual components is reasonably described, but how they fit together (and how they relate to the specific questions addressed) could be more clearly described.

We thank the reviewer for this comment. To better explain how each step fit together we have added a few paragraphs in the beginning of the Methods section. The included text reads (Line 112 in the revised manuscript): 

“The method used in this paper consists of two main parts. The first part is to set up and run experiments in an IBM for each of the two cities we investigate. The second part is to estimate the change, and the uncertainty, in the effective reproduction number resulting from school opening. 

The model setup consists of two parts. First, we needed to construct the network on which we simulate transmission, and secondly, we needed to specify the transmission model. 

Constructing the transmission network was a non-trivial task. Since we aimed to evaluate the effect of school opening, we needed to include schools in the network. We also needed to include families in the network since the families tie together different schools. For instance, if two siblings attend two different schools, their family represents a connection between the two schools that can cause an outbreak to spread from one school to another. Our approach was first to construct families randomly using probability distributions consistent with the demographic data for each city. The random assignment of schools used probabilities derived from the municipalities' data for the number of students in each school. If two children in the same family both attended middle school, they were assigned to the same school. The same was true for elementary schools and kindergartens. Before high school, Norwegian children attend schools in their neighborhoods. To capture this structure, we first assigned children to middle schools and then distributed younger siblings (and younger children without older siblings) to elementary schools and kindergartens in middle schools' neighborhoods. 

Once the networks were constructed, we modeled disease transmission using a standard probabilistic IBM with three layers. In this model, each individual's probability of being infected was proportional to a weighted sum over all infectious individuals in the city. We used one weight for family members, one for schoolmates, and one for other individuals that the person could encounter. The transmission rates in families, in schools, and for other encounters were the most critical parameters in the model. 

We ran model experiments with varying parameters, and for each parameter set, we carried out two experiments. One where schools remained closed, i.e., the transmission rates in schools were set to zero, and a second experiment where the schools were opened. We estimated the effective reproduction number for each pair of runs and compared the value with schools closed to the value estimated from the run where the schools were opened. To estimate the reproduction number, we fitted the data from the model runs to a susceptible-exposed-infected-recovery (SEIR) model. The rationale for using an SEIR model was that this technique resembles the method used to assess Norway's reproduction number at the time. 

The final part of our analysis was to analyze the change in the reproduction number over the set of model experiments. We focused on the dependence between the difference in the reproduction number (between schools closed and schools opened) and the transmission rate in schools, which was the most critical and most uncertain transmission model. 

The details of our methodology are described below.” 

Further comments below:

please number pages and lines!

Sorry for not including line numbers and page numbers. We have added them to the revised manuscript. 

check spelling & punctuation: "serios" somewhere, and misplaced commas.

Thanks. We have corrected grammar and spelling mistakes throughout the manuscript. 

Introduction:

"no other study uses as detailed data as we have collected in our experiments" Are you referring to empirical data or model outputs here?

We were referring to modelling set-ups and the level of specific details on schools. However, the sentence is unnecessary, and we have removed it from the revised manuscript. 

Obviously this is a very fast moving area, one further study I am aware of (though not involved with) that has appeared since is https://doi.org/10.1016/S2352-4642(20)30250-9

Thanks. We have added a reference to this work and rephrased this paragraph. It now reads (Line 101 in the revised manuscript): 

“There exist similar IBM studies that focus on the effects of NPIs on the spread of SARS-CoV-2 [26]. However, as far as we know, no other studies that evaluated a controlled school reopening using this methodology were published when this work was carried out in the spring of 2020. However, the field is evolving rapidly, and in August, an IBM study on strategy for reopening schools in the UK was published [27]. We use a novel approach to infer results that remain robust under the uncertainties about SARS-CoV-2 virus dynamics.” 

Methods:

Seriously infected children make a reduced contribution to the school-based force of infection, based on the assumption that they will stay home from school. Do these children make an increased contribution to the home-based force of infection (where they are now spending more time)?

In the model, seriously infected children are more infectious (the factor 1+c), but not in schools. I agree with the reviewer that it would be better to explicitly describe an increased contribution from symptomatic children spending more time at home. This can be incorporated in future work. 

Do you assume any change in behaviour (eg, isolation) relating to development of symptomatic/severe disease in other age groups? How might this influence your interpretation of model behaviour?

This is not explicitly built into the model, isolation of symptomatic individuals is thought to be an important measure to suppress the spread in Norway in the spring of 2020. In the model, the effect of these measures are is built into the transmission coefficients. Since we kept beta1 and beta3 constant we assumed that isolation and quarantine practices for other age groups remained unchanged as schools opened. It is clear that our interpretation of model results assumes that behavior in other age groups remained unchanged. We have added this to the discussion. The sentence at end of the first paragraph in the discussion now reads (Line 460 in the revised manuscript): 

“Our findings suggest that it is challenging to infer these data through assessments of countrywide school openings, even if one makes the unrealistic assumption that the change in R after April 20 is not influenced by other factors than schools opening, such as changes in behavior in other age groups.” 

The timing of school opening and closures is a unclear: "In the model experiments, schools remain closed for 14 days (β 2 = 0), and then kindergartens opened. Grades 1-4 opened one week later." When schools closed, was this all schools, including kindergartens? After Grades 1-4 opened, did higher grades also open? It may be useful when describing model to associate which school years / age groups / birth years are associated with which type of school, and use consistent terminology throughout.

We have clarified this and expanded the first paragraph under “Model runs and analyses”. This paragraph now reads (Line 249 in the revised manuscript): 

“In the IBM we carried out a number of simulations with transmission coefficients (β_1, β_2, and β_3), each drawn randomly from uniform distributions on the interval 0.2-1. In the model experiments, all schools and kindergartens were closed during the first 14 days of the simulation (β_2=0). These 14 days correspond to the period from April 6 to April 20. During this period all kindergartens and schools in Norway were closed. Following the actual implementation of school reopening, we opened kindergartens (children born 2014 or lager) after running the models for 14 days and grades 1-4 (children born 2010-2013) opened after 21 days. The reopening of schools for older children coincided with other changes in restrictions and was not included in this modelling study. “

How were the numbers of experiments (121 for Oslo and 295 for Tromso) chosen? Were these "experiments" independent stochastic realisations using the same parameter values, or were the parameter values resampled for every experiment? For the comparison results, did each of these experiments involve running the simulation twice (ie, with and without school opening)? Please explain this more clearly.

The modelling was carried out in the days leading up to reopening of schools and the number of experiments were those that had been carried out by that time. Our assessment was that running more additional experiments would not reduce uncertainty significantly, as the main uncertainty is associated with the unknown transmission rates. 

We agree that the description of the runs was unclear. We have edited the sentences. They now read (Line 270 in the revised manuscript): 

“For the city of Oslo, we carried out 121 pairs of model experiments, and for the city of Tromsø we carried out 295 pairs. The two runs in a pair were identical for the first 14 days, and after that they branched in two directions: one run where schools opened, and a second run with the same parameters, that continued with schools closed. Model parameters were resampled for each pair.”

I am a bit confused by the description of model runs. In the first paragraph it suggests that you sampled each of the three beta parameters from uniform distributions. The followin paragraph suggests that beta_2 was fixed to 0.94 following Ferguson's models.

Beta_2 was not fixed. We just normalized it so that r=1 corresponded to 0.94 following Ferguson’s models. To make this clear we added the following sentences to the text (Line 288 in the revised manuscript): 

“We varied β_2 in the range from 0.2 per day to 1.0 per day, i.e., r between 0.21 and 1.06. The transmission rates β_1and β_3 were also chosen from the uniform distribution on the interval (0.2,1).”

Am I understanding correctly that the sole purpose of the compartmental model was to enable estimation of R from hospitalisation data, that the IBM could then be fit to? It would be helpful to explain this more clearly, as it took some piecing together.

We agree that this could be stated clearer. We added the following sentences early in the methods section (Line 159 in the revised manuscript): 

“To estimate the reproduction number, we fitted the data from the model runs to a susceptible-exposed-infected-recovery (SEIR) model. The rationale for using an SEIR model was that this technique resembled the method used to assess Norway's reproduction number at the time.” 

Figure 2: it would be helpful to label the rows / columns with identifying factors (eg, city / weight parameters) rather than relying on the caption.

The cities where already labeled in the figure. We have updated it so that it includes the other information. 

Supp mat: references to Eurosurveillance would need to be updated.

Ups. This is now removed. 

Reviewer #2

thank you for the attempt to investigate the effectiveness of closing of schools during this pandemic. First, it is best to carefully check your English - there exists some spelling mistakes and grammatical errors; also, please check on how to cite reference when quoting. 

Thanks. We have corrected grammar and spelling mistakes throughout the manuscript. 

Second, you have mentioned on page 9 that you have used "ensembles of model experiments" of which I could not find the discussions on this. Can you clarify? 

We simply mean the set of model runs with different parameter values. To clarify we have rephrased the sentence. It now reads (Line 411 in the revised manuscript): 

“We found the relationship between ΔR and r to be approximately linear over the ensembles set of simulations with different parameter.”

Third, will it be possible to inform on what basis does the parameter c were assigned such values? also, can you justify the assigned values for psi and alpha? 

The psi and alpha are unchanged from the influenza models as we had no data to support assigning other values. The age-dependence in the c_i-s are, of course, very uncertain. And we don’t really know precisely what severe disease means in the model, since it is also relative to the threshold for staying home from school. To clarify these points, we added the following text under Supplementary Table 1 (Line 10 in the revised Supplementary Material): 

“The probability of serious infection is highly uncertain, and we do not know the threshold for staying home from school during the pandemic. We used values that incorporate a strong age-dependence consistent with recent studies [1, 2], and can account for pre-symptomatic and asymptomatic transmission between children. Parameters Ψ and α are not changed from the influenza-version of the model [3].” 

1. Davies NG, Klepac P, Liu Y, Prem K, Jit M, Pearson CAB, et al. Age-dependent effects in the transmission and control of COVID-19 epidemics. Nature Medicine. 2020;26(8):1205-11. doi: 10.1038/s41591-020-0962-9.

2. Ludvigsson JF. Systematic review of COVID-19 in children shows milder cases and a better prognosis than adults. Acta Paediatrica. 2020;109(6):1088-95. doi: https://doi.org/10.1111/apa.15270.

3. Ferguson NM, Cummings DAT, Cauchemez S, Fraser C, Riley S, Meeyai A, et al. Strategies for containing an emerging influenza pandemic in Southeast Asia. Nature. 2005;437(7056):209-14. doi: 10.1038/nature04017.

Fourth, on the R estimation, it was not clear why were there two sets of systems of equations, one on SEI and the other on SEIR? Are the lamda different from the SEIR model to the ones defined on the main text? 

We use one “simple” SEIR model to estimate R from in the model runs. There is no need for something more complex here, because the time series I(t) is available in the model results. We use a slightly more complex model for the real-world R estimate, which was based on hospitalizations, and needed more compartments. This model was constructed and calibrated by the Norwegian Institute of Public Health (NIPH), and we just simplified it from a metapopulation model for each municipality, to a one-population version. To clarify this, we have included the following text (Line 29 in the revised Supplementary Material): 

“To estimate the basic reproduction number in Norway we used a one-population version of the SEIR model used by the NIPH [4]. This model was slightly more complex than the model above, which we used to estimate R from experiments of the IBM. The reason for the difference is that the time series of the number of infectious was not available, and in the spring of 2020, the R-estimates in Norway were based on the number of hospitalized patients. We used the same parameter values as in [4] , which are consistent with the ones used in the simpler SEIR model described above.“ 

and, which of the parameters that you have introduced may be in relation to the social/physical distancing? 

In all models the parameters that mostly determine R, and hence the effect of social distancing, are the transmission rates (the betas). 

Also, will it be possible to explain further on the contact matrix estimated from the cellphone data. Why can't contact matrix from Prem et al be used? 

Our understanding is that, early in the pandemic, the NIPH (of Norway) adapted an influenza model developed as a PhD project at the University of Oslo. This influenza model was designed for Bangladesh where Grameenphone, a partner of the Norwegian telecommunication company Telenor, is the largest cellphone provider. The model was implemented for Norway, using weekly updated contact matrices. These matrices include average movement between municipalities, but nothing else. 

In the paper we stated clearly that we did not use contact matrices or cell phone data. Hence, we do not want to expand on this point in the manuscript. 

Finally, would you say that your findings differs from that of Zhang's ?

This is a good question, but we are hesitant to comment too strongly on the work of Zhang et al. Our work considers only reopening of schools for children born 2010 or later, so we cannot compare directly. As far as we can see, our results are not inconsistent with the findings of Zhang et al. 

Reviewer #3: 

In the manuscripts, the authors considered the effect of kindergartens and schools opening on COVID-19 dynamics in Norway. Both basic and effective reproduction numbers were estimated relative schools openings via the individual-based model (IBM). The found little relative change in reproduction number from opening the schools and concluded that controlled school opening might not alter the cause of progress made on COVID-19 in many countries considering this option. The methodology and results are well-written and supported by good discussion and conclusion. In general, the manuscript is well-organized, and my comments on the work are below:

Major comments:

The authors follow the stochastic model for influenza directly in [1] for COVID-19. The dynamics of COVID-19 is entirely differently from influenza [2]. I can see that some of the parameters are updated for COVID-19, but pre-symptomatic transmission and asymptomatic are two of the factors that make COVID-19 unique. These are not accounted for in this modelling. Thus, I wonder if this stochastic simulation captures the essential dynamics of COVID-19.

We agree that one can question if the model captures the essential dynamics of COVID-19, but we must keep in mind that the question addressed is the effect of school opening in Norway at a time when a “lock down” had suppressed the spread of the disease. Hence, for this particular study, we have to ask if the model captures pre-symptomatic and asymptomatic transmission in schools after reopening. This is partially incorporated in the model through the terms c_i, and the term Psi, which model the effect of absence from schools because of symptoms. We have updated our model for COVID-19 by introducing a strong age dependence in the parameters c_i. To make this clearer we have added the following text below Supplementary Table 1 (Line 10 in the revised Supplementary Material): 

“The probability of serious infection is highly uncertain, and we do not know the threshold for staying home from school during the pandemic. We used values that incorporate a strong age-dependence consistent with recent studies [1, 2], and can account for pre-symptomatic and asymptomatic transmission between children. Parameters Ψ and α are not changed from the influenza-version of the model [3].” 

It has been shown that there exist age-dependent susceptibility and infectivity in COVID-19 dynamics [3]. More of this assertion comes from a population with more younger age group than older age group vs with a population with more older age group than younger age group. The Norway age-structure is pyramid [4], meaning there are fewer young children than adults. Thus, allowing contacts between children and adults via school re-opening may not significantly impact COVID-19. However, all these need to be adjusted for in the modelling. How does infectiousness and susceptibility of younger children differ from adults in this modelling? Note, your c_j and ρ_j may answer relative infectiousness but not relative susceptibility.

The relative susceptibility enters the model through the transmission rates. Since we only used one beta for schools (beta_2), this implies that we could only incorporate a coarse age-dependence in susceptibility (children born 2010 and later, versus the rest of the population). Nevertheless, the set-up can model lower susceptibility in children through the beta_2 coefficient. The importance and uncertainty of susceptibility in children is one of the reasons that we did not use a fixed beta_2 value, but instead presented results for a range of different values. 

As mention in 1 above, COVID-19 has its dynamics. Thus, a SIR or SEIR compartmental like models may be over-simplification of reality. The authors did not provide the compartmental like nature of their IBM. Hence, it is easier to assume it is SIR. Also, what is the essence of using SEIR model to estimate R_0? The IBM model can be fitted to data to estimate this. How does the dynamics of your SEEIIR model differ from IBM. Since Age is considered, I guessed age-structured compartmental model with appropriate contact matrix would be closed the approximation of the IBM model.

The reason we used a SEIR model to estimate R in model runs was that we wanted direct comparability to the estimates the NIPH were carried for Norway in the spring of 2020. These estimates were carried out with the SEEIIR model that we describe. The SEEIIR model is convenient when estimating R from hospitalization data, but an SEIR model will suffice for the model runs where the time series I(t) are available. We agree that an age-structured compartmental model is a good alternative to an IBM for studying school opening. 

To clarify how and why the compartmental models are used we added the following text to the Supplementary Material (Line 29 in the revised Supplementary Material): 

“To estimate the basic reproduction number in Norway we used a one-population version of the SEIR model used by the NIPH [4]. This model was slightly more complex than the model above, which we used to estimate R from experiments of the IBM. The reason for the difference is that the time series of the number of infectious was not available, and in the spring of 2020, the R-estimates in Norway were based on the number of hospitalized patients. We used the same parameter values as in [4], which are consistent with the ones used in the simpler SEIR model described above.“ 

The results are a little bit difficult to follow. For instance, “For each city, the figures are based on the whole ensemble of simulations but weighted by a probability density p(R) for the reproduction number prior to April 20,…”. p(R) is not defined anywhere else, even in the supplementary! I may miss this.

Similar to 4 above, “ the Gaussian weight distribution is centred around R=0.6, 0.8 and 1.0, …”. Which Gaussian distribution? All you have in figure 2 are straight lines of the basic reproduction numbers. “In panel B, the probability of having R+ΔR>1 after school opening in Oslo is estimated be 0.13…”. How? Nothing in the methodology pointed to this, and even in panel E, figure 2, what you have are straight lines. Maybe a better figure that depicts these changes will be better.

We have made this part clearer by adding descriptions into the figure. In addition, we have expanded the description. We have removed the notation p(R) as it may be confusing. 

The paragraph now reads (Line 351 in the revised manuscript): 

“This risk of a new wave of infections depends on the magnitude of R prior to school opening. As will be shown below, we found that if in-school transmission rates for the SARS-CoV-2 virus are similar to what is believed to be the case for influenza pandemics (r=1), we estimated for Oslo ΔR=0.10 (95%CI 0.04-0.16) and for Tromsø, ΔR=0.14 (95%CI 0.01-0.25). This means that the most interesting range of R prior to school opening to explore is the interval 0.6-1.0. The result of such an exploration is shown in Figure 2. The three panels show the estimated R before April 20 and after April 27, assuming different values of R before April 20. The way we were able to assume different initial values of R, was that we weighted simulations by their estimated R values before April 20 using gaussian weight distributions centered around R=0.6, 0.8, and 1.0, respectively, with standard deviation 0.1. To the right in each panel, the R-values after school opening is plotted. The most interesting case is probably R centered around 0.8 (panels B and E), because this is close to the most probable R estimated for the two cities (see Supplementary Figure 1), and in this case, the weighted sum over model runs with R greater than 1 after April 27 gave a probability of 0.13 for having R+ΔR>1 after school opening in Oslo. In Tromsø, this probability is somewhat higher, 0.28, as shown in panel E. Since the magnitude of R prior to school opening is only known with considerable uncertainty (Supplementary Figure 1), these probabilities are also associated with great uncertainty, but their magnitudes still provide an indication of the risk for a second wave of infection associated with school openings.” 

Minor comments:

Change “serios” to “serious”

“Form” to “are” in the model runs and analyses section

Thanks. These and other mistakes are corrected.

---

## [Decision Letter · Decision Letter 1]

9 Feb 2021

Modelling Suggests Limited Change in the Reproduction Number from Reopening Norwegian Kindergartens and Schools During the COVID-19 Pandemic

PONE-D-20-25739R1

Dear Dr. Rypdal,

We’re pleased to inform you that your manuscript has been judged scientifically suitable for publication and will be formally accepted for publication once it meets all outstanding technical requirements.

Kind regards,

Eric HY Lau, Ph.D.

Academic Editor

PLOS ONE

Additional Editor Comments (optional):

Reviewers' comments:

Reviewer's Responses to Questions

**Comments to the Author**

1. If the authors have adequately addressed your comments raised in a previous round of review and you feel that this manuscript is now acceptable for publication, you may indicate that here to bypass the “Comments to the Author” section, enter your conflict of interest statement in the “Confidential to Editor” section, and submit your "Accept" recommendation.

Reviewer #1: All comments have been addressed

Reviewer #2: All comments have been addressed

Reviewer #3: All comments have been addressed

2. Is the manuscript technically sound, and do the data support the conclusions?

Reviewer #1: Yes

Reviewer #2: Yes

Reviewer #3: Yes

3. Has the statistical analysis been performed appropriately and rigorously? 

Reviewer #1: N/A

Reviewer #2: Yes

Reviewer #3: Yes

4. Have the authors made all data underlying the findings in their manuscript fully available?

Reviewer #1: No

Reviewer #2: Yes

Reviewer #3: Yes

5. Is the manuscript presented in an intelligible fashion and written in standard English?

Reviewer #1: Yes

Reviewer #2: Yes

Reviewer #3: Yes

6. Review Comments to the Author

Reviewer #1: Thank you for your attention to comments. The revised paper reads much more clearly. One minor comment; the new sentence: line 61: "The reopening of schools on April 20 and April 27 were not accompanied by relaxation of other control measures until later in the spring of 2020." is ungrammatical, and would read better as: "The reopening of schools on April 20 and April 27 was not accompanied by relaxation of other control measures, which occurred later in the spring of 2020."

Reviewer #2: Thank you for addressing all the concerns put forward earlier. I am satisfied with your response. The paper reads better now and I believe the paper will be beneficial specifically to disease modelling community.

Reviewer #3: (No Response)

7. PLOS authors have the option to publish the peer review history of their article (what does this mean?). If published, this will include your full peer review and any attached files.

Reviewer #1: No

Reviewer #2: No

Reviewer #3: No

---

## [Editor Report · Acceptance letter]

16 Feb 2021

PONE-D-20-25739R1 

Modelling Suggests Limited Change in the Reproduction Number from Reopening Norwegian Kindergartens and Schools During the COVID-19 Pandemic 

Dear Dr. Rypdal:

I'm pleased to inform you that your manuscript has been deemed suitable for publication in PLOS ONE. Congratulations! Your manuscript is now with our production department. 

Kind regards, 

on behalf of

Dr. Eric HY Lau 

Academic Editor

PLOS ONE